# Further Evidence on Import Demand Function and Income Inequality

**Ioanna Konstantakopoulou** 

Centre of Planning and Economic Research, 11 Amerikis Street, 106 72 Athens, Greece; ik@kepe.gr;
Tel.: +30-210-3676406

**Abstract:** In advanced economies, rising inequality has become a significant economic issue. Our paper examines one dimension of the impact of inequality. This study employs panel estimators that tackle heterogeneity and cross-sectional dependence to estimate the impact of income inequality on import demand. In addition, we use a Bayesian approach to the cointegrated VAR model as well as a model that allows for stochastic trends and cross-sectional dependence. Annual panel data for the period from 1995 to 2016 on OECD countries are used. The empirical results show that inequality has a positive and significant effect on import demand. The estimation also yields some other expected results, viz. that the income and price elasticity of import demand function are positive and negative, respectively.

**Keywords:** import demand; income inequality; cross-sectional dependence; system GMM; panel cointegrated VAR

---

## 1. Introduction

Inequality matters for each economy. Recent works (World Inequality Report 2018; Solt 2016) have underlined the growing gap between the rich and the poor in developed countries, raising concerns for policymakers. This trend has various economic implications and is affected by various economic variables. In our work, we focus on the trade variable and especially on imports.

The literature that links trade and income inequality has received a great deal of attention over the years. The two main dimensions of research focus on whether trade affects income inequality, and vice versa. The former relationship has been examined by an extensive number of studies (Richardson 1995; Panagariya 2000; Chakrabarti 2000; Goldberg and Pavcnik 2007; Bergh and Nilsson 2010; Jaumotte et al. 2013; Roser and Cuaresma 2016; Lin and Fu 2016; Lim and McNelis 2016; Mahadevan et al. 2017; Anderson 2020; Hirte et al. 2020; Furusawa et al. 2020), while there is far less literature covering the way in which income inequality affects trade (Francois and Kaplan 1996; Mitra and Trindade 2005; Dalgin et al. 2008; Fajgelbaumy et al. 2011; Bekkers et al. 2012; Asteriou et al. 2014; Michaels et al. 2014; Tridico 2018; Hummels and Lee 2018).

We examine the latter relationship and focus on one aspect of the subject, namely, the impact of income inequality on import demand, thus taking into account the demand-side effects. To investigate the research question of our paper, we rely on the following works by Katsimi and Moutos (2011) and Adam et al. (2012), who examined the effects of changes in inequality on import demand. The theoretical explanation of a link between inequality and trade patterns is based on non-homothetic preferences in vertically differentiated products, as income distribution affects the demand for products

of different qualities (Linder 1961[1]; Thursby and Thursby 1987; Fajgelbaumy et al. 2011). In particular, the elasticity of demand for each good depends not only on income but also on the distribution of that income or per capita income, when preferences are non-homothetic. This assumption is different from the view of traditional trade theory, which suggests identical and homothetic preferences across all countries.

The dependence of trade flows not only on aggregate income but also on aggregate income distribution implies that trade patterns vary across countries. We assume that preferences are non-homothetic, i.e., that some goods are "luxury goods" and some are "necessity goods." Countries with income elasticity (poor countries) smaller than 1 import goods and prefer to consume "necessity goods," while countries with income elasticity larger than 1 (rich countries) prefer more complex and "luxury goods." Therefore, we expect that the impact of income inequality on import demand will vary between countries depending on the quality of the goods they trade. If the quality of the goods is "luxury," we expect that an increase in income inequality (if we assume that the income of the rich is increased by one amount and the income of the poor is reduced by the same amount) will produce a positive effect on the import demand of the importing country. On the contrary, if the goods are "necessity goods," a corresponding increase would have a negative effect on import demand when the examined county is rich. In our empirical analysis, we use a sample of high-income countries, so we expect to have a positive sign for the elasticity of income inequality. The way we choose to prove the above relationship by assessing an "enriched" import demand function.

Trade elasticities have been analyzed and are widely used in the international empirical literature. Assessing the impact of inequality on import demand is achieved through estimating elasticities. Several studies have dealt with the estimation of the import demand function using various determinant variables, different data samples and countries, and different econometric methodologies. In particular, Giovannetti (1989); Giansoldati and Gregori (2017); Konstantakopoulou (2018) consider the different final expenditure components and relative prices to examine their effects on imports. The two latter studies use panel data analysis based on a sample of 33 OECD (Organization for Economic Co-operation and Development) countries and Euro area countries. Senhadji (1998) estimates the elasticities of the import demand function, using an appropriate variable to capture income: GDP minus exports. He uses a time series analysis for a sample of 66 countries. The estimated short-run price and income elasticity result have the expected signs. The findings of this study reveal that industrial countries have both higher income and lower price elasticities than developing countries. Gafar (1988) and Sinha (1997) confirm the importance of relative prices and real income in determining import demand. Tang (2003) uses four definitions of domestic activity—namely, gross domestic product (GDP), GDP minus exports, national cash flow, and final expenditure components—to investigate the impact of these domestic activity measures on China's import demand. He finds that the price elasticity for China's aggregate import demand is inelastic. Harb (2005) uses GDP and GDP minus exports as determinant variables to estimate the elasticity of the import demand function, using a heterogeneous panel of developed and developing countries. He shows that income elasticities in developing countries are not different from unity, on average, and are higher than in developed countries. Caporale and Chui (1999) estimate the income and price elasticity of trade across 21 countries using a more recent time-series technique. Gozgor (2014) shows that the driving factor of import demand is economic growth estimating aggregate and disaggregate function for China.

Gregori and Giansoldati (2020) estimated both world- and country-specific elasticities using the import intensity-adjusted demand measure. They show that the worldwide elasticity of import intensity-adjusted demand is close to one, leading to the conclusion that, to estimate import demand, we should also include intermediate goods.

---

[1]    Linder's hypothesis assumes that firms in any country produce products tailored to the dominant preferences of local consumers and sell them globally to others who share these tastes.

Their main finding indicates that faster growing economies have lower income demand elasticities for their imports. Katsimi and Moutos (2011) examine the effects of changes in inequality on import demand using US data. They find that inequality significantly affects the demand for imports, except for the imports of services where the impact of inequality is ambiguous. Adam et al. (2012) assess the impact of changes in income inequality on the demand for imports using panel data for 59 developing and developed countries. They find that the effects are positive for high-income countries and negative for low-income countries.

The main objective of this paper is to re-examine the impact of inequality on the import demand function using more recent econometric methods and extend the empirical literature of Adam et al. (2012) and Katsimi and Moutos (2011). We use several estimators to deal with heterogeneity and cross-sectional dependence and ensure robust results. Thus, we obtain robust results on the basis of the panel fully modified ordinary least square (FMOLS) estimator and the dynamic common correlated effects pooled mean group estimator (CCE-PMG). We account for cross-sectional dependence and allow for heterogeneity, and we use the Bai et al. (2009) estimator that allows for stochastic trends (stationary or not) as well as cross-sectional dependence. Our empirical procedure is divided into two parts. The first part includes panel cointegration analysis and estimation; the second part is based on panel cointegrated VAR methodology. The cointegration analysis testing for the existence of a long-run equilibrium relationship between our variables allows us to estimate the long-run coefficients of our estimated model. This procedure ensures that no spurious regression arises any longer. The estimation of the panel cointegrated VAR model using the Bayesian approach to cointegration has two advantages. First, it can produce whole probability distributions for each parameter that are valid for any sample size. Second, it permits the identification of the VAR model with more accuracy. Our findings indicate a significant impact of income inequality on import demand; this is verified using all estimation methods and the panel cointegration VAR methodology. Moreover, the income and price elasticity of the import demand function is positive and negative, respectively.

This paper makes a two-fold contribution to the existing literature: First, the empirical methodology applied has not been implemented in other works estimating the import demand function. Second, the estimation of the import demand function will help policymakers to comprehend better the effect of variables causing changes to the trade balance and deal with any external imbalances. The import demand function's estimated coefficients will show the extent of the change caused to import demand by a percentage change in determinant variables. In addition, incorporating the income inequality variable into the import demand function will assist policymakers in implementing stabilization policies for external sector deficits, while harmonizing their social policy accordingly. Furthermore, trade flows depend on relative prices; thus, an appropriate trade policy using the estimated coefficients of the import demand function might be the best response to preventing external imbalances. The determination of the variables that affect import demand may help policymakers predict and effectively deal with various external disturbances.

The paper is structured as follows. Section 2 presents data. Section 3 develops the theoretical framework and econometric methodology. Our empirical results are reported in Section 4. Section 5 presents the conclusion and policy implications arising from our results.

## 2. Data Description

We proxy inequality using the Gini coefficient of disposable income of households (post-tax, post transfer) comes from the Standardized World Income Inequality Database (SWIID) version 6.2 of Solt (2009, 2016). The Gini index evaluates the extent to which the distribution of income within an economy deviates from a perfect normal distribution. Imports are the imports of goods and services at constant 2010 USD, which refer to the market size of countries (Adam et al. 2012). The real GDP variable is GDP at constant 2010 USD. Real Imports and real GDP are supplied by the World Bank's World Development Indicators (WDI). The relative prices variable is defined as the ratio of import

prices to GDP deflator. We proxy the external competitiveness using the relative prices variable (Katsimi and Moutos 2011). Import prices, and GDP deflator come from the WDI of the World Bank.

We estimate the import demand function over a balanced panel at annual frequency between 1995 and 2016, comprising the following 36 OECD countries: Australia, Austria, Belgium, Canada, Chile, Czech Republic, Denmark, Estonia, Finland, France, Germany, Greece, Hungary, Iceland, Ireland, Israel, Italy, Japan, Korea, Latvia, Lithuania, Luxembourg, Mexico, the Netherlands, New Zealand, Norway, Poland, Portugal, Slovak Republic, Slovenia, Spain, Sweden, Switzerland, Turkey, the United Kingdom, and the United States. The time period of our sample and the countries of our analysis have been selected based on the data availability of the variables we use in our empirical analysis.

Table 1 presents the descriptive statistics of the sample variables in the 36 OECD countries, during the full sample period (1995–2016). The descriptive statistics indicate that the real GDP variable has the highest standard deviation of 1.622, while inequality has the lowest mean of 0.308. Real GDP has the highest mean of 26.57, while the inequality variable has the lowest standard deviation of 0.056.

**Table 1.** Descriptive statistics for the period 1995–2016.

| Basic Descriptive Statistics | | | | |
|---|---|---|---|---|
| **Variables** | **Mean** | **Std. Dev.** | **Min.** | **Max.** |
| Real Imports | 22.572 | 1.371 | 21.985 | 28.706 |
| Real GDP | 26.57 | 1.622 | 22.842 | 30.462 |
| Inequality | 0.308 | 0.056 | 0.217 | 0.485 |
| Imports prices | 0.314 | 0.409 | −0.799 | 2.47 |

Note. Descriptive statistics: Mean is mean. Std. Dev. is standard deviation. Min is minimum. Max is maximum.

## 3. Theoretical Framework and Empirical Methods

### 3.1. Aggregate Import Demand Model

The traditional import demand function relies on the imperfect substitutes model, based on a representative household's maximization of utility function subject to a budget constraint. In this theoretical model, real imports are related to real output and relative prices (Goldstein and Khan 1985). We modify the model by adding income inequality to take into account the effects of inequality on trade (Adam et al. 2012; Katsimi and Moutos 2011). Thus, income inequality is one of the determinant variables of the import demand function. Based on this framework, we consider a heterogeneous panel regression model,

$$logimp_{it} = \gamma_0 + \gamma_1 logy_{it} + \gamma_2 logrp_{it} + \gamma_3 logineq_{it} + \varepsilon_{it} \tag{1}$$

where $i = 1, \ldots N$, denotes county $i$ and $t$ denotes time, $imp_{it}$ is the nature logarithm real imports in country $i$ at time $t$, $y_{it}$ is the nature logarithm real GDP, $rp_{it}$ denotes the nature logarithm import prices, and $ineq_{it}$ is the nature logarithm income inequality. We measure the sensitivity of import demand to changes in income, income inequality and price variables by estimating Equation (1). The income elasticity of demand for imports is expected to be positive, while price elasticity is expected to be negative. Moreover, according to the theoretical foundation, we expect the effect of income inequality on import demand to be positive. The justification of this positive sign comes from the hypothesis of vertically differentiated products and non-homothetic preferences.

### 3.2. Empirical Methodology

Our econometric analysis is conducted as follows. In part I, we test for cross section dependence across the countries. Second, we test whether there is a long run equilibrium relationship between the variables, using appropriate panel cointegration tests. Then, having confirmed that the variables are cointegrated, we estimate the long-run elasticities using the following methods: (i) the panel fully

modified OLS estimator (FMOLS), and (ii) the dynamic common correlated effects pooled mean group estimator (CCE-PMG). In part II, we implement the panel cointegrated VAR methodology.

### 3.2.1. Panel Analysis

#### Testing for Cross-Sectional Dependence (CD)

In this study, we use Pesaran's CD test (Pesaran 2004) to control for cross-sectional dependence. The CD test is based on average of pair-wise correlation coefficients of OLS residuals from individual regressions in the panel model. The CD test statistic is as follows:

$$CD = \sqrt{\frac{2T}{N(N-1)}} \left( \sum_{i=1}^{N-1} \sum_{j=i+1}^{N} \hat{\rho}_{ij} \right) \tag{2}$$

where $\hat{\rho}_{ij}$ is the sample estimate of the pair-wise correlation of the residuals, $\hat{\rho}_{ji} = \dfrac{\sum_{t=1}^{T} \hat{e}_{it}\hat{e}_{jt}}{\left( \sum_{t=1}^{T} (\hat{e}_{it})^2 \right)^{\frac{1}{2}} \left( \sum_{t=1}^{T} (\hat{e}_{jt})^2 \right)^{\frac{1}{2}}}$,
$N$ is the number of cross-sections, and $T$ is the time dimension. The CD statistic follows a standard normal distribution under the null of cross-sectional independence.

#### Panel Unit Root Test

We use a second-generation panel unit root test (PURT) that is known as CIPS (cross-sectionally augmented IPS) test, proposed by Pesaran (2007). This test allows for cross-sectional dependence in the data. Pesaran augments the standard DF (or ADF) regressions with the cross-sectional averages of lagged levels and first-differences of the individual series in the panel to solve the cross-dependence problem. For this purpose, he used the following cross-sectional augmented Dickey–Fuller (CADF) regression:

$$\Delta y_{it} = \alpha_i + \rho_i y_{it-1} + \beta_i \widetilde{y}_{t-1} + \sum_{j=0}^{n} \gamma_{ij} \Delta \widetilde{y}_{t-1} + \sum_{j=0}^{n} \delta_{ij} y_{it-1} + e_{it} \tag{3}$$

A cross-sectionally augmented version of the IPS test is defined, as follows:

$$CIPS(N,T) = \left( \frac{1}{N} \right) \sum_{i=1}^{N} t_i(N,\ T) \tag{4}$$

where $t_i(N,T)$ is the cross-sectionally augmented Dickey–Fuller statistic for the $i$th cross-section unit given by the $t$-ratio of the coefficient of $y_{i,t-1}$, in the CADF regression defined by (3). Pesaran (2007) has determined the critical values of CIPS for different deterministic terms.

#### Panel Cointegration Tests

The following step of econometric analysis is to test whether imports, output, relative prices, and income inequality are cointegrated using several panel cointegration tests (Pedroni 1999, 2004; Westerlund 2007). Pedroni's tests allow for heterogeneity in the intercepts intercept and slopes coefficients of the cointegrating equation. There are seven tests with four being within-dimension (panel) and three being between-dimension (group). In particular, he constructs three non-parametric tests that correct for serial correlation (panel v-stat, panel rho-stat, and panel PP-stat) and a fourth parametric test similar to the ADF-type test (panel ADF-stat). These panel statistics are based on pooling he data along the within dimension of the panel. The other three cointegration statistics are based on a group mean approach (group rho-stat, group PP-stat, and group ADF-stat). Pedroni's statistics are based on the estimated residuals from the panel cointegration following regression:

$$y_{it} = \alpha_i + \beta_i t + \gamma'_i X_{it} + e_{it} \tag{5}$$

For a time series panel of observables $y_{it}$ and $X_{it}$ for units $i = 1 \ldots \ldots, N$ over time $t = 1 \ldots \ldots, T$, $X_{it}$ is a $(1 \times k)$ vector for each units $i$, $\beta_i$ is a scalar and $\gamma'_i$ is a $(k \times 1)$ vector for each units $i$. The variables $y_{it}$ and $X_{it}$ are assumed to be integrated of order one, for each unit $i$ of the panel, and under the null of no cointegration. The estimated residual be specified as follows:

$$e_{it} = \rho_i e_{it-1} + u_{it}$$

The null hypothesis of all seven tests is absence of cointegration. Pedroni's statistics are normally distributed.

We also use a second-generation panel cointegration test proposed by Westerlund (2007). Westerlund (2007) developed four test statistics to handle cross-sectional dependence through bootstrapping to test the null of no cointegration. These panel cointegration tests have satisfactory small-sample properties and high power relative to residual-based tests proposed by Pedroni ( 2004). In addition, Westerlund calculates asymptotic and bootstrap p-values, making inference possible under very general forms of cross-sectional dependence. These statistics are differed as to the alternative hypotheses assumed. Two tests (panel tests, $P_t - stat$ and $P_\alpha - stat$) are designed to test the alternative hypothesis that the panel is cointegrated as a whole, while the other two test (group mean tests $G_t - stat$ and $G_a - stat$) the alternative that there is at least one individual that is cointegrated. All tests are normally distributed. Each test accommodates individual-specific short-run dynamics, serially correlated error terms and non-strictly exogenous regressors, individual-specific intercept and trend terms, and individual-specific slope parameters.

Estimation

We implement the FMOLS method appropriate for heterogeneous cointegrated panels that, as Pedroni (2000) noted, are associated with the fact that a standard panel OLS estimator is asymptotically biased and its distribution is dependent on nuisance parameters associated with the underlying dynamic processes. To eliminate bias due to endogeneity of regressors, Pedroni developed the group-means FMOLS estimator by incorporating semi-parametric correction into the OLS estimator. The technique accounts fully for heterogeneity in short-run dynamics as well as for fixed effects. Consider the following cointegrated system for a simple two variable panel of $i = 1 \ldots \ldots, N$ over units,

$$y_{it} = \alpha_{it} + \beta x_{it} + u_{it} \tag{6}$$

The FMOLS estimator is

$$\hat{\beta}_{i,FMOLS} = N^{-1} \sum_{i=1}^{N} \left( \sum_{t=1}^{T} (x_{it} - \bar{x}_i)^2 \right)^{-1} \left( \sum_{t=1}^{T} (x_{it} - \bar{x}_i) y_{it}^* - T\hat{y}_i \right)$$

where $y_{it}^* = (x_{it} - \bar{x}_i) - \frac{\hat{\Omega}_{21i}}{\hat{\Omega}_{22i}} \Delta x_{it}$, $\hat{y}_{it} = \hat{\Gamma}_{21i} - \hat{\Omega}_{21i}^0 - \frac{\hat{\Omega}_{21i}}{\hat{\Omega}_{22i}} (\hat{\Gamma}_{22i} - \hat{\Omega}_{22i}^0)$, $\hat{\Omega}$ and $\hat{\Gamma}$ are covariances and sums of autocovariances obtained from the long-run covariance matrix for model (6).

We also use the CCEPMG estimator of Chudik and Pesaran (2015), who extend the common correlated effects (CCE, Pesaran 2006) approach to dynamic heterogeneous panel data models with weakly exogenous regressors. This method addresses the cross-country heterogeneity, cross-sectional dependence, and possible unit roots in factors. Inequality differs across cross-section counties and mainly depends on specific unobserved factors that vary across countries. They show that the CCE mean group estimator continues to be valid but the following two conditions must be satisfied to deal with the dynamics: a sufficient number of lags of cross-section averages must be included in individual equations of the panel, and the number of cross-section averages must be at least as large as the number of unobserved common factors.

Bai et al. (2009) propose two iterative procedures that estimate the slope parameters as well as the stochastic trends. The estimators are known as CupBC (continuously updated and bias-corrected) and CupFM (continuously updated and fully modified). Bai et al. (2009) establish consistency and derive asymptotic distributions. The two estimators allow inference to be conducted using standard test statistics. As they mention, "The estimators are also valid when there are mixed stationary and non-stationary factors, as well as when the factors are all stationary" (p. 82).

### 3.2.2. Panel Cointegrated VAR

Let $y_t$ be a $n \times 1$ vector time series which has a cointegrated VAR representation with $r$ cointegrating vectors:

$$\Delta y_t = \sum_{l=1}^{L} \Gamma_l \Delta y_{t-l} + \Pi y_{t-1} + \Phi d_t + \varepsilon_t, \tag{7}$$

where $\varepsilon_t \sim iidN_t(0, \Sigma)$, $\Pi = \alpha\beta'\Pi$, $\alpha, \beta$ are $n \times r$, $\Gamma_l$ are $n \times r$ $(l = 1, \ldots, L)$, $\Phi$ is $n \times d$ and $d_t$ is $m \times 1$ vector containing deterministic variables. The model can be written as

$$Y_t = \Gamma X_{2t} + \alpha\beta' X_{1t} + \Phi d_t + \varepsilon_t, \tag{8}$$

where $\Gamma = [\Gamma_1, \ldots, \Gamma_k]$, $Y_t = \Delta x_t$, $X_{1t} = y_{t-1}$, $X_{2t} = [\Delta y_{t-1}, \ldots, \Delta y_{t-k}]'$. In different notation,

$$Y = \Gamma X_2 + \alpha\beta' X_1 + \Phi d + \varepsilon, \tag{9}$$

after collecting all observations. We assume

$$Ey_t\left(\varepsilon_{i,t}\, \varepsilon'_{j,\,s}\right) = \Sigma_{ij}, \text{ for } t = s \text{ and zero otherwise} \tag{10}$$

We allow for arbitrary cross-sectional correlation. Define $y = \left(y'_1, \ldots y'_N\right)'$, $b = \left(b'_1, \ldots b'_N\right)'$ and a matrix $x$ whose diagonal is $(x_1, \ldots, x_N)$. The system is $y = xb + e$, and the likelihood function is

$$\begin{aligned} L(b, \Sigma, \beta) &= \left|\Sigma\right|_{-T/2} \exp\left\{-\tfrac{1}{2}(y - xb)' V_e^{-1}(y - xb)\right\} \\ &= \Sigma\Big|_{-T/2} \exp\left\{-\tfrac{1}{2}\left[s^2 + (b - \hat{b})' V^{-1}(b - \hat{b})\right]\right\}, \end{aligned} \tag{11}$$

where $s^2 = y'My$, $M = V_e^{-1} - V_e^{-1}xVx'V_e^{-1}$, $\hat{b} = Vx'V_e^{-1}y$, $V = (x'V_e^{-1}x)^{-1}$, $V_e = \Sigma \otimes I_T$. With a flat or normal prior on $b$, the conditional (on $\Sigma$ and $\beta$) posterior of $b$ is normal.

The normalization $\beta = [I_r MB']'$ does not solve the local and global identification problems of the vector error correction model (VECM). Several authors, including Strachan (2003), Strachan and Inder (2004), and Villani (2005, 2006) propose various approaches and Koop et al. (2009) extend the framework of Strachan and Inder (2004) to the panel cointegration model. Strachan and Inder (2004) use $\beta'\beta = I_r$ to identify the cointegrating vectors without restrictions on the cointegrating space. In the VECM, only the space spanned by the columns of $\beta$ is identifiable, and we only have information on $P = span(\beta)$ which belongs to the Grassmann manifold $G_{n,r}$, i.e., the space of all $r-$dimensional planes of $R^n$. A flat prior for the cointegration space is therefore given by the uniform distribution placed on $G_{n,r}$. A proper prior that can be used is the "semi-orthogonal prior":

$$p(b, \Sigma, b_\beta) \propto \left|\Sigma\right|_{-(Nn+1)/2}, \; \beta'_i \beta_i = 1. \tag{12}$$

Koop et al. (2009) discuss another prior which imposes soft homogeneity (similarity) constraints on the elements of the cointegrating vectors across units:

$$b \sim N(0, h^{-1}\underline{V}) \tag{13}$$

where $h$ is a scalar which controls the degrees of informativeness or the precision of the prior. We use this approach and to specify $h$ and the elements of $\underline{V}$ we proceed as in Koop et al. (2009).

## 4. Empirical Results

In this section, we report our empirical findings. The results of the CD (Cross-sectional Dependence) tests (Table 2) strongly reject the null hypothesis for all variables, and the mean correlation coefficients are large in absolute value. The results for the unit root tests are reported in Table 3, for the series in levels and first differences. The results of the CIPS test for lag orders $p = 0$; 1; 2 (Table 3) indicate that the variables in first differences are stationary, that is, most statistics reject the null. Table 4 shows the rejection of the null hypothesis of no cointegration by four test statistics, implying that a long-run relationship between the variables does exist. Table 5 presents the results of the Westerlund (2007) cointegration test for cross-sectional dependence. The results also provide sufficient evidence of cointegration for the variables, that is, the null hypothesis is rejected by the three statistics.

**Table 2.** Cross-sectional dependence tests.

| Variables | CD-Statistic | Abs. Values Corr. Coefficient |
|---|---|---|
| Real Imports | 66.47 *** | 0.580 |
| Real GDP | 63.53 *** | 0.553 |
| Inequality | 5.98 *** | 0.537 |
| Imports prices | 86.06 *** | 0.748 |

*** indicates rejection of the null hypothesis at the 1% level of significance.

**Table 3.** Cross-sectionally augmented IPS (CIPS) panel unit root tests (PURTs).

| | Real GDP | Real Imports | Inequality | Imports Prices |
|---|---|---|---|---|
| | | Level | | |
| $p = 0$ | −2.027 | −2.078 | −1.867 | −2.029 |
| $p = 1$ | −2.492 | −2.432 | −2.210 | −2.459 |
| | | First Difference | | |
| $p = 0$ | −3.124 *** | −3.434 *** | −3.125 *** | −3.874 *** |
| $p = 1$ | −2.895 ** | −2.686 *** | −2.594 ** | 3.264 *** |

*** and ** indicate rejection of the null hypothesis at the 1% and 5% level of significance, respectively.

**Table 4.** Pedroni's (2004) results.

| | Panel Cointegration Statistics | Group-Mean Panel Cointegration Statistics |
|---|---|---|
| Variance ratio | −1.064 | |
| PP rho-statistics | 2.105 | 3.095 |
| PP t-statistics | −4.818 *** | −5.117 *** |
| ADF t-statistics | −5.460 *** | −5.843 *** |

*** indicates rejection of null hypothesis at the 1% level of significance. Lags are selected according to SBC.

**Table 5.** Westerlund's (2007) results.

| Statistics | Value | Z-Value | Robust $p$ Value |
|---|---|---|---|
| $G_T$ | −2.756 *** | −2.870 | 0.008 |
| $G_\alpha$ | −5.968 | −4.733 | 0.887 |
| $P_T$ | −13.089 *** | −5.112 | 0.000 |
| $P_\alpha$ | −6.231 ** | −3.945 | 0.026 |

*** and ** indicates rejection of the null hypothesis at 1% and 5% levels of significance, respectively. 800 bootstrap replications are used to correct for cross-sectional dependence.

*Estimation Results*

As we find evidence of a long-run equilibrium relationship among real imports, real GDP, inequality, and import prices, we proceed to estimating the long run coefficients of our model. We present the estimated results of the fully modified ordinary least square, FMOLS and Common correlated effects pooled mean group, CCEPMG, as well as the results from the Bayesian panel cointegration in Table 6. Bayesian estimation indicated that all coefficients are significant. The elasticity of imports with respect to inequality is 0.282 (0.145) with the Bayesian method (FMOLS), implying that, in the long run, a one percent increase in inequality is associated with an increase in imports of goods and services by 0.282% (0.145%). The estimated coefficients of the CCEPMG estimator are also presented in Table 6. The results of the CCEPMG estimator clearly suggest that inequality is positively associated with import demand and is statistically significant. Moreover, the estimated coefficient of real GDP is highly significant and positive. This means that OECD countries tend to increase imports as their real GDP increases. The coefficient of relative prices is statistically significant and negative, as expected. Table 7 presents the CupBC and CupFM estimators that reinforce the results of all estimates of the paper confirming the positive effect of real GDP, the negative effect of import prices and the positive effect on import demand on inequality.

**Table 6.** Estimation results Fully Modified Ordinary Least Square (FMOLS), Bayesian, and Common correlated effects pooled mean group (CCEPMG).

| | FMOLS | Bayesian Panel Cointegrated VAR | CCEPMG | 95% C.I. |
|---|---|---|---|---|
| **Dependent Variable: Real Imports** | | | | |
| Real GDP | 1.685 *** (52.361) | 1.532 *** (26.54) | 1.904 *** (17.22) | (1.342 1.942) |
| Inequality | 0.145 *** (8.322) | 0.282 *** (6.398) | 0.242 *** (2.725) | (−0.005 0.217) |
| Imports prices | −0.155 *** (−6.653) | −0.168 *** (−7.936) | −0.159 ** (−1.894) | (−0.197 0.020) |

Note: *** and ** at the 1% and 5% levels of statistical significance, respectively. All estimations include a constant country specific term. Lags are selected according to the Schwarz Bayesian criterion (SBC). *t*-values are in parentheses. C.I.: confidence interval. For the Bayesian panel cointegration results, the ratio of posterior mean to posterior s.d. is reported in parentheses.

**Table 7.** Results with cross-sectional dependence (Bai and Kao 2006; Bai et al. 2009).

| Dependent Variable: Real Imports | | |
|---|---|---|
| | **CupBC** | **CupFM** |
| Real GDP | 1.235 *** (37.12) | 1.212 *** (22.76) |
| Inequality | 0.177 *** (7.554) | 0.251 *** (11.34) |
| Imports prices | −0.143 *** (−12.65) | −0.122 *** (−15.31) |

Note: *** at the 1% level of statistical significance, respectively. All estimations include a constant country specific term. Lags are selected according to the Schwarz Bayesian criterion (SBC). *t*-values are in parentheses. C.I.: confidence interval. For the Bayesian panel cointegration results, the ratio of posterior mean to posterior s.d. is reported in parentheses.

## 5. Conclusions

In this paper, we estimated an import demand function, taking into consideration the impact of income inequality on import demand for 36 OECD countries. This study employs panel estimators that tackle heterogeneity and cross-sectional dependence to estimate the impact of income inequality on import demand.

Our paper leads to several interesting results. All of the estimated coefficients have signs compatible with the theoretical arguments in empirical literature. In particular, the elasticities of income and prices are positive and negative, respectively; in line with previous studies (Senhadji 1998; Gafar 1988; Sinha 1997; Gozgor 2014).

Our estimates suggest a significant positive impact of income inequality on import demand (Katsimi and Moutos 2011; Hummels and Lee 2018). This confirms the theoretical basis of the model where in high-income countries an increase in income inequality leads to an increase in import demand. Income inequality is a crucial fraction of trade balance.

The results have several policy implications. First, real GDP has a highly significant and highly elastic impact on import demand. This means that a 1% decrease in real GDP will cause an even smaller reduction of import demand implication, which will lead to an improvement of the trade balance. On the contrary, a corresponding increase will have a negative effect on the current account and will cause concern and mobilize economic policymakers.

Second, the finding of a positive relationship between income inequality and import demand will help policymakers better predict and manage trade balance imbalances. In addition, incorporating the income inequality variable into the import demand function will help policymakers in implementing stabilization policies for external sector deficits while harmonizing their social policy accordingly. Policies for reduction of income inequality should lead to an improvement of the trade balance and a smoothing out of potential social outbursts.

Finally, minor relative price changes do not have a powerful effect on import demand; it is suggested that substantial relative price swings are necessary in order to produce a considerable reallocation of trade flows. The estimated price elasticity is quite low, implying that trade policy appears to be a weak tool in the hands of policymakers. Changes in import prices will have little impact on import demand, and consequently on the trade balance of OECD countries.

**Funding:** This research received no external funding.

**Conflicts of Interest:** The authors declare no conflict of interest.

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
