# Peer review of "Further Evidence on Import Demand Function and Income Inequality"

_economies, doi:10.3390/economies8040091_

Round 1

Reviewer 1 Report

Summary: The manuscript analyzes the impact of income inequality in import demand which is demand side-effect study. The unit root tests present evidence of long-run relationship among variables. Therefore, the manuscript uses a Bayesian approach to the cointegrated VAR model.  The results show that income inequality has a positive effect on import demand.  

Evaluation: The manuscript is not appropriate for publication and need to be modified. The method part is not easy to understand for readers and need to be improved. Furthermore, all references should be added into the method part since they are taken from time series analysis.The equation functions are not clear to follow them in the manuscript.The results are not complete and need to be explained more in details.The results do not present policy implications.There are many grammatical errors in manuscript and need to be modified.

Author Response

Author's Reply to the Review Report (Reviewer 1)

Comments and Suggestions for Authors

Summary: The manuscript analyzes the impact of income inequality in import demand which is demand side-effect study. The unit root tests present evidence of long-run relationship among variables. Therefore, the manuscript uses a Bayesian approach to the cointegrated VAR model.  The results show that income inequality has a positive effect on import demand.  

Evaluation: The manuscript is not appropriate for publication and need to be modified. The method part is not easy to understand for readers and need to be improved. Furthermore, all references should be added into the method part since they are taken from time series analysis.The equation functions are not clear to follow them in the manuscript.The results are not complete and need to be explained more in details.The results do not present policy implications. There are many grammatical errors in manuscript and need to be modified.

Answer:

Following the referee’s comments, we have more clearly and deeply analyzed the policy implications in conclusions. In addition, we have carefully read our article to correct any grammatical errors.

Moreover, we would like to mention that this study employs panel data analysis and not time series analysis.

Following the referee’s comment, we have enriched the introduction with the appropriate bibliography to make it more complete. The following are the additional bibliographic references.

Asteriou, D., Dimelis, S., Moudatsou, A., 2014. Globalization and income inequality: A panel data econometric approach for the EU27 countries. Economic Modelling, 36, 592-599.

Bekkers, E., Francois, J., Manchin, M., 2012. Import prices, income, and inequality. European Economic Review, 56 (4): 848-869.

Gafar, J. S., 1988. The determinants of import demand in Trinidad and Tobago: 1967–84. Applied Economics, 20 (3): 303-313.

Goldberg, P. K., Pavcnik, N., 2007. Distributional Effects of Globalization in Developing Countries. Journal of Economic Literature, 45 (1), 39-82.

 Gozgor, G., 2014. Aggregated and disaggregated import demand in China: An empirical study. Economic Modelling, 43, 1-8.

Jaumotte, F., Lall, S., Papageorgiou, C., 2013. Rising Income Inequality: Technology, or Trade and Financial Globalization?. IMF Economic Review, 61, 271–309.

Lin, F., Fu, D., 2016. Trade, Institution Quality and Income Inequality. World Development, 77, 129-142.

Richardson, J. D., 1995. Income inequality and trade: how to think, what to conclude. Journal of economic perspectives, 9 (3): 33-55.

Sinha, D., 1997. Determinants of import demand in Thailand. International Economic Journal, 11 (4): 73-873.

Tridico, P., 2018. The determinants of income inequality in OECD countries. Cambridge Journal of Economics, 42: 1009-1042.

Reviewer 2 Report

This paper examines the the impact of income inequality on import demand for OECD countries employing Bayesian approach to the cointegrated VAR model to the data spanning from 1995 to 2016. The study finds that inequality has a positive and significant effect on import demand. In my view, this paper contributes to literature as it represents a group of countries. However, the paper may not be accepted unless the following issues are fully addressed.

  1. I recommend the authors to extend the literature review.
  2. Descriptive statistics and charts of the used variables in analysis are missing. These would provide an insightful overview of the variables and their development over time.
  3. The authors correctly review many studies and cite many methods implemented in the literature. However, they do not explain why they prefer panel cointegration methods to other methodologies.

Author Response

Author's Reply to the Review Report (Reviewer 2)

Comments and Suggestions for Authors

This paper examines the the impact of income inequality on import demand for OECD countries employing Bayesian approach to the cointegrated VAR model to the data spanning from 1995 to 2016. The study finds that inequality has a positive and significant effect on import demand. In my view, this paper contributes to literature as it represents a group of countries. However, the paper may not be accepted unless the following issues are fully addressed.

  1. I recommend the authors to extend the literature review.

 Answer:

Following the referee’s comments, we have enriched the introduction with the appropriate bibliography to make it more complete. The following are the additional bibliographic references.

Asteriou, D., Dimelis, S., Moudatsou, A., 2014. Globalization and income inequality: A panel data econometric approach for the EU27 countries. Economic Modelling, 36: 592-599.

Bekkers, E., Francois, J., Manchin, M., 2012. Import prices, income, and inequality. European Economic Review, 56 (4): 848-869.

Gafar, J. S., 1988. The determinants of import demand in Trinidad and Tobago: 1967–84. Applied Economics, 20 (3): 303-313.

Goldberg, P. K., Pavcnik, N., 2007. Distributional Effects of Globalization in Developing Countries. Journal of Economic Literature, 45 (1): 39-82.

 Gozgor, G., 2014. Aggregated and disaggregated import demand in China: An empirical study. Economic Modelling, 43: 1-8.

Jaumotte, F., Lall, S., Papageorgiou, C., 2013. Rising Income Inequality: Technology, or Trade and Financial Globalization?. IMF Economic Review, 61: 271–309.

Lin, F., Fu, D., 2016. Trade, Institution Quality and Income Inequality. World Development, 77: 129-142.

Richardson, J. D., 1995. Income inequality and trade: how to think, what to conclude. Journal of economic perspectives, 9 (3): 33-55.

Sinha, D., 1997. Determinants of import demand in Thailand. International Economic Journal, 11 (4): 73-873.

Tridico, P., 2018. The determinants of income inequality in OECD countries. Cambridge Journal of Economics, 42: 1009-1042.

  1. Descriptive statistics and charts of the used variables in analysis are missing. These would provide an insightful overview of the variables and their development over time.

Answer:

Following the referee’s comments, we have calculated the descriptive statistics of variables in our model. Table 1 provides a summary of the descriptive statistics. We have also added paragraph 3 to section 3, which analyses the statistics of the variables.

  1. The authors correctly review many studies and cite many methods implemented in the literature. However, they do not explain why they prefer panel cointegration methods to other methodologies.

Answer:

Answer: Following the referee’s comments, we have analyzed why we select cointegration analysis in our paper. Thus, we have added the explanation in rows 99 to 102.

Reviewer 3 Report

This is an interesting study.

Some improvements are necessary in terms of making the results robust.

In the introduction, the paper uses the term "society and economy". In which way are they different? Also note that Solt (2016) is not a report, but a research paper, so please consider rephrasing the sentence.

The paper uses reduced from model to examine the determinants of import demand, with factors like GDP, relative prices (import prices and GDP deflator) and income inequality measured by GINI coefficient. How robust is the result of this study? This should be examined with factors like the domestic prices and import prices taken separately in the model, GDP per capita, and the variables noted in the literature like Giansoldati and Gregori (2017) or Konstantakopoulou (2018) which included variables like consumption, investment, government expenditure, exports. This can be shown to support your main findings regarding income inequality.

In the conclusion section, the study argues that "Changes in the exchange rate will have little impact on import demand...". However, exchange rate was not tested in the model, so how can this claim be made? It would be useful to reconsider this statement or examine the effect of exchange rate in the model.

How close is the paper's finding to the existing studies in the literature? At the moment, the paper does not discuss how the results agree/disagree with existing literature.

In addition to the results table, please also provide tables on descriptive statistics and correlation matrix of grouped sample.

The policy aspects need to be improved, and also revised to make them easier to follow, especially paragraph 3 of the conclusion section. 

What are some limitations of the paper? The study is only focused on OECD countries which are majorly developed countries, hence the findings cannot be generalized to countries outside OECD group and especially small and developing economies. 

Some additional points

  • Table 5 has the coefficient for Import price under CCEPMG as negative and the t-ratio is positive. This is not consistent.
  • Table 6 does not clearly show the "Import Price" results 

Please review, use insights from and cite the following papers accordingly. These studies are relevant to the topic of the paper.

  • Bekkers, E., Francois, J., & Manchin, M. (2012). Import prices, income, and inequality. European Economic Review, 56(4), 848-869.
  • Richardson, J. D. (1995). Income inequality and trade: how to think, what to conclude. Journal of economic perspectives, 9(3), 33-55.
  • Tridico, P. (2018). The determinants of income inequality in OECD countries. Cambridge Journal of Economics, 42(4), 1009-1042.
  • Gafar, J. S. (1988). The determinants of import demand in Trinidad and Tobago: 1967–84. Applied Economics, 20(3), 303-313.
  • Sinha, D. (1997). Determinants of import demand in Thailand. International Economic Journal, 11(4), 73-873.

All the best.

Author Response

Comments and Suggestions for Authors

This is an interesting study.

Some improvements are necessary in terms of making the results robust.

In the introduction, the paper uses the term "society and economy". In which way are they different? Also note that Solt (2016) is not a report, but a research paper, so please consider rephrasing the sentence.

Answer:

We have rewritten the two sentences following the referee’s instructions.

The paper uses reduced from model to examine the determinants of import demand, with factors like GDP, relative prices (import prices and GDP deflator) and income inequality measured by GINI coefficient. How robust is the result of this study? This should be examined with factors like the domestic prices and import prices taken separately in the model, GDP per capita, and the variables noted in the literature like Giansoldati and Gregori (2017) or Konstantakopoulou (2018) which included variables like consumption, investment, government expenditure, exports. This can be shown to support your main findings regarding income inequality.

Answer:

This is an interesting research question for future consideration. As for the robustness of our results, we have applied several estimators so as to ensure robust results.

In the conclusion section, the study argues that "Changes in the exchange rate will have little impact on import demand...". However, exchange rate was not tested in the model, so how can this claim be made? It would be useful to reconsider this statement or examine the effect of exchange rate in the model.

Answer:

Following the referee’s comments, we have made the required change. The variable estimated is import prices and not the exchange rate as erroneously written.

How close is the paper's finding to the existing studies in the literature? At the moment, the paper does not discuss how the results agree/disagree with existing literature.

Answer:

Following the referee’s comment, we have rewritten the conclusions of the paper focusing on findings in relation to other studies and the policy implications.

In addition to the results table, please also provide tables on descriptive statistics and correlation matrix of grouped sample.

Answer:

Following the referee’s comments, we have calculated the descriptive statistics of variables of our model. Table 1 provides a summary of the descriptive statistics. In section 3, we added paragraph 3, which analyses the statistics of the variables.

The policy aspects need to be improved, and also revised to make them easier to follow, especially paragraph 3 of the conclusion section. 

Answer:

Following the referee’s comment, we have clarified the policy implications of our paper. Thus, we have made significant changes in the conclusions.

What are some limitations of the paper? The study is only focused on OECD countries which are majorly developed countries, hence the findings cannot be generalized to countries outside OECD group and especially small and developing economies. 

Answer:

The limited availability of data did not allow us to consider our research question for small and developing countries as well. We also note this indirectly in section where we state that: The selection of the time period of our sample as well as the selected countries that make up the two panels of our analysis have been carried out based on the data availability of the variables we use in our empirical analysis.

Some additional points

  • Table 5 has the coefficient for Import price under CCEPMG as negative and the t-ratio is positive. This is not consistent.

Answer:

Following the referee’s comment, we have made the necessary correction.

  • Table 6 does not clearly show the "Import Price" results 

Answer:

Following the comment of the referee, we have made all the necessary corrections.

Please review, use insights from and cite the following papers accordingly. These studies are relevant to the topic of the paper.

  • Bekkers, E., Francois, J., & Manchin, M. (2012). Import prices, income, and inequality. European Economic Review, 56(4), 848-869.
  • Richardson, J. D. (1995). Income inequality and trade: how to think, what to conclude.Journal of economic perspectives, 9(3), 33-55.
  • Tridico, P. (2018). The determinants of income inequality in OECD countries. Cambridge Journal of Economics, 42(4), 1009-1042.
  • Gafar, J. S. (1988). The determinants of import demand in Trinidad and Tobago: 1967–84. Applied Economics, 20(3), 303-313.
  • Sinha, D. (1997). Determinants of import demand in Thailand. International Economic Journal, 11(4), 73-873.

All the best.

Answer:

Following the referee’s comment, we have added all the suggested references in the introduction of our paper.

Round 2

Reviewer 1 Report

Summary: The manuscript analyzes the impact of income inequality in import demand which is demand side-effect study. The unit root tests present evidence of long-run relationship among variables. Therefore, the manuscript uses a Bayesian approach to the cointegrated VAR model. The results show that income inequality has a positive effect on import demand.  

Evaluation: There is no any difference between the revised version of the manuscript with previous version. The revised version of the manuscript is not appropriate for publication and need to be modified. Again, I see the same issues as before. The method part is not easy to understand for readers and need to be improved. In the revised version, a few references added in the text and reference section. Again, the equation functions are not clear to follow them in the manuscript, and should be reorganized. In the revised version, the results section is as the same before without adding any explanation. The results are not complete and need to be explained more in details.The results do not present policy implications.The manuscript needs to be modified because of English.

The authors answered that this study employs panel data analysis and not time series analysis which is not correct. The authors are using a balanced panel at annual frequency between 1995 and 2016 which is panel data time series analysis. I suggest that the authors read the "panel data time series analysis". The authors used cointegration VAR method for panel data which is a time series analysis by using panel data. The authors  have used pooled panel time series analysis without providing any explanation. Why they used Pooled method and not fixed effect or random effect model? In the manuscript, there is not any answer to these question. The contribution of this research is not strong.

Reviewer 3 Report

The revised paper shows improvements. 

I have no further comments to make.

Author Response

I have not replied to the Reviewer 3, because he has noted the following Comments and Suggestions for Authors: “The revised paper shows improvements. I have no further comments to make.” Therefore, I do not think I have to reply to reviewer 3.